# Reactivation of Vertebral Growth Plate Function in Vertebral Body Tethering in an Animal Model

**DOI:** 10.3390/ijms231911596

**Published:** 2022-09-30

**Authors:** Michał Latalski, Tomasz Szponder, Grzegorz Starobrat, Edward Warda, Magdalena Wójciak, Sławomir Dresler, Anna Danielewicz, Jan Sawicki, Ireneusz Sowa

**Affiliations:** 1Childrens’ Orthopedic Department, Medical University of Lublin, Gębali 6, 20-093 Lublin, Poland; 2Department and Clinic of Animal Surgery, Faculty of Veterinary Medicine, University of Life Sciences, Głęboka 30, 20-612 Lublin, Poland; 3Orthopedics and Traumatology Department, University Hospital, Jaczewskiego 8, 20-954 Lublin, Poland; 4Department of Analytical Chemistry, Medical University of Lublin, Chodzki 4a, 20-093 Lublin, Poland; 5Department of Plant Physiology and Biophysics, Institute of Biological Science, Maria Curie-Skłodowska University, 19 Akademicka Str., 20-033 Lublin, Poland

**Keywords:** vertebral body tethering, scoliosis, growth-friendly techniques, growth plate, chondrocytes, spine, growth

## Abstract

Flexible spine tethering is a relatively novel fusionless surgical technique that aims to correct scoliosis based on growth modulation due to the pressure exerted on the vertebral body epiphyseal growth plate. The correction occurs in two phases: immediate intraoperative and postoperative with growth. The aim of this study was to evaluate the reactivation of vertebral growth plate function after applying corrective forces. The rat tail model was used. Asymmetric compression and distraction of caudal growth plates were performed using a modified external fixation apparatus. Radiological and histopathological data were analysed. After three weeks of correction, the activity of the structures increased across the entire growth plate width, and the plate was thickened. The height of the hypertrophic layer and chondrocytes on the concave side doubled in height. The height of chondrocytes and the cartilage thickness on the concave and central sides after the correction did not differ statistically significantly from the control group. Initiation of the correction of scoliosis in the growing spine, with relief of the pressure on the growth plate, allows the return of the physiological activity of the growth cartilage and restoration of the deformed vertebral body.

## 1. Introduction

The thoracolumbosacral orthosis is the current standard of care for skeletally immature patients with idiopathic scoliosis (IS) with moderate curves (between 20° and 40°) [1]. Bracing may prevent curve progression in some patients but rarely results in a reduction in the deformity. Many curves, especially in young patients, may ultimately progress and result in traditional surgical intervention with a spinal fusion. Fusionless approaches are emerging for patients with remaining growth potential and in need of a surgical treatment [2]. Flexible spine tethering is a relatively novel fusionless surgical technique that aims to correct scoliotic deformities based on growth modulation due to the pressure exerted on the vertebral body epiphyseal growth plate [3]. It is dedicated as a non-fusion surgical procedure for the correction of scoliosis in skeletally immature patients. Candidates for anterior vertebral body tethering (AVBT) are patients with at least 1 year of growth remaining with moderate scoliosis. The curve correction occurs in two phases: immediate intraoperative and postoperative with growth. Immediate correction is achieved through the patient positioning (placement of the patient in the lateral decubitus position) and intraoperative techniques—the reduction is achieved with manual tension applied through the cord. The second phase of correction appears with spinal growth. Some authors suggest that creating compression on the convex side of the vertebra slows growth on the convex side and facilitates growth on the concave side of the spine [4]. Ohashi et al. used the rat ulna loading model and reported that high loads led to an increase in the growth plate height and the hypertrophic height, probably due to damage to the growth plate that prevented the mineralisation of the hypertrophic zone into bone [5]. The researchers speculate that if the rats were followed for an extended period after cessation of loading, the growth plates might be observed to return to normal endochondral ossification. However, there is not enough literature regarding an assessment of the reactivation of vertebral growth plate function after applying forces correcting deformation. The aim of the current study was to evaluate this topic.

## 2. Results

The apparatus was well tolerated by the animals, and the rats were able to control tail movements in the caudal and cephalic parts from the apparatus despite the stabilisation of the internal vertebrae.

### 2.1. Radiological Examination

The radiological measurements showed that during the 6 weeks of the experiment, the mean increase in the vertebra length in the control group (control vertebrae) was 625 µm (velocity 15 µm/day), which is 9.5% of the body length. The mean increase in the epiphysis width in this period was 250 µm (6 µm/day). After 3 weeks, all the animals exhibited a wedge-shaped deformity in the compressed vertebrae (groups II and III); next, the spine axis was improved in all animals subjected to decompression. Detailed data are shown in Figure 1.

### 2.2. Histopathological Examination

Three parameters were measured: the thickness of the growth plate and the hypertrophic layer, and the chondrocyte layer height. The results are shown in Figure 2. Representative histopathological images are shown in Figure 3, Figure 4 and Figure 5.

A comparison of the histopathological images of the cartilage in the control group K 1 with the vertebrae of the control animals K3 after the end of the experiment indicates a noticeable reduction in the thickness of the growth plate and the hypertrophic layer and a simultaneous increase in the chondrocyte height, which is a typical phenomenon in the proper development of the backbone.

The height of the hypertrophic layer in the control group decreased after the first three weeks of the experiment and was unchanged until the end of the study. The width of the growth plate decreased with the age of the animals. The greatest decline in the width was recorded after the first three weeks of the experiment (by 28.1% of the original value). A further 15.3% decline was noted after the next 21 days. These differences were statistically significant.

In comparison with the control group, the histopathological image of the cartilage from the concave, central, and convex sides of the body and the vertebrae of the control animals after 73 days of the experiment (completion of scoliosis induction) showed a reduced width of the entire growth plate in the induced scoliosis (Figure 4a), with the most extensive narrowing on the concave side versus the central and convex sides. The chondrocyte height was significantly lower in the deformed vertebra body than in the control group (Figure 4c). It decreased with the closer distance to the concave side. In the experimental group, the height of the hypertrophic layer (Figure 4b) was lower than in the control, and its largest reduction was visible on the compression side.

After three weeks of regeneration, the activity of the structures increased across the entire growth plate width, and the plate was thickened. The height of the hypertrophic layer and the chondrocytes on the concave side doubled (Figure 5b,c). After the end of the experiment, the height of the chondrocytes (Figure 5c) across the entire cartilage width did not differ from that in the control group. The cartilage thickness on the concave and central sides after the regeneration did not differ statistically significantly from that in the control group.

None of the examined layers on the convex side, where the distraction forces were applied, increased in a statistically significant manner compared to the control group.

The PCA results (Figure 6) indicated that the first two components explained 87% of the total variance. The first PC was largely positively loaded by cartilage thickness and hypertrophic layer thickness, while the chondrocyte height variables were partially positively correlated with PC1 and negatively correlated with PC2. The first PC facilitated the separation of individuals with 3 weeks until scoliosis induction from the rest of the samples. Additionally, it was found that these observations were weakly positively correlated with PC2. Most observations, including the control in the third and sixth weeks of the experiment and scoliosis after regeneration (scoliosis 6), are located in the centre of the plot and under axis 1 (negative correlation with PC2). In contrast, the control individuals at the start of the experiment exhibited a positive correlation with both PC1 and PC2.

## 3. Discussion

Various investigators have experimented with spinal growth modulation in animal models [6,7]. Anterior vertebral body tethering is a novel growth-modulating surgical technique for the correction of spinal deformity in patients with adolescent scoliosis [3]. 

It is a well-known fact that forces acting on the bone element can increase or decrease the velocity of its growth. Thus, the mechanobiological phenomenon can influence skeletal growth and development. 

Regardless of the aetiology of the curvature, there is a biological and mechanical relationship with the degree of deformation. The development of mathematical models and analyses of vertebral body growth were mainly carried out by Bick. The author found that the growth of the spine takes place in the vertebral boundary plate and is similar to the growth of the epiphyseal zones of long bones [8]. The Hueter–Volkmann law is known in the biology of bone growth. Formulated in the 19th century but still frequently used in medical literature, it assumes that increased pressure acting on the growth plate retards bone growth; conversely, reduced pressure or even tension accelerates it [9,10,11]. However, this statement is purely qualitative and does not take into account the rate of the load rise/fall, the frequency and amplitude of these changes, the dimensions of the growth plate, the degree of its maturity, and growth potential. These variables may have a different effect on the sensitivity to mechanical stress. However, the growth of the spine slightly depends on physiological dynamic loads. Niehoff et al. have shown that the differences in vertebral growth in rats showing different levels of activity are small [12]. Additionally, the forces occurring in humans at the level of intervertebral discs amount to approx. 300 N and 700–1100 N in a sitting position. In dynamic loads, this value can be as high as 2 to 14 kN depending on gender, age, and body weight [13,14,15]. It is obvious that in a growing skeleton, the loads acting on the growth cartilage have different values and directions. Additionally, the cartilages themselves also have different shapes. It is indisputable that bone formation and growth take place in complex mechanical and geometric conditions. The complex bone structure, especially the heterogeneity of the growth plate, is also an important element [16]. Carter and Wong formulated a theory that facilitates a biomechanical analysis of the growing bone, taking into account the triaxial state of stresses occurring in the growth plate [17,18,19,20]. They relied on the Sine criteria used to predict the initiation of fatigue fractures in metals [21]. Stevens et al. supplemented this theory with the observations reported by Mikic et al., who found that the mechanobiological contribution to the cartilage maturation rate is a linear combination of the maximum octahedral shear stress and the minimum hydrostatic stress in a region of cartilage during a complete loading cycle [22,23]. Piszczatowski reported the non-uniformity of mechanical stimulation of the growth cartilage within its volume, despite the uniform axial loading acting on the growth plate. His research confirms the importance of the growth plate inhomogeneity, although the analyses were focused rather on the inner structure of the growth cartilage, while neglecting the role of the bone plates and the fibrous ring [16]. However, in addition to the material aspect, the geometry of the growth plate (shape, relation between height and diameter) may be important [24]. Piszczatowski also proved that the direction of the force has a significant impact on the model of mechanical stimuli occurring within the growth cartilage. As suggested by the author, bone growth is therefore dependent on the direction of the load. The shape of the growth cartilage itself, in this context, is of secondary importance. However, the location where the load acts on the growth cartilage remains important [16,24]. 

A more complex analysis of the mechanical influences on the bone growth was presented by FROST [25]. The author proposed the “Chondral Growth–Force Response characteristics” (CGFRC), where the mechanical stimulation of the bone growth was dependent not only on the sense of stresses but also on their value. For stresses not exceeding the physiological range, the endochondral growth runs faster in the case of compression compared to tension. 

According to this dependence, the increasing pressure causes the growth to accelerate until the limit value of the compressive force is not exceeded. Then, the activity slows down to blockage of the growth cartilage [26]. Rajasekaran et al. analysed radiographs of 234 spine segments in 63 children with kyphosis due to tuberculosis of the spine. The follow-up was 15 years. As shown in their research, under the influence of stretching and compression forces, the growth may accelerate, both locally and in the entire border-line plate, slow down, stop completely, or even destroy the vertebrae [27]. Growth modulation was observed in adjacent segments within the kyphotic region, which were uninvolved in the lesion. In children with less than 30° kyphosis, the corresponding facet joints were intact and the minimal increase in unique loads was seen to accelerate the growth, which partly compensated for the deformity. When the deformity exceeded 40°, suppression of growth was observed, which led to a decrease in the growth of the anterior parts of the vertebral endplates. This resulted in anteriorly wedged vertebrae in 53 segments. These changes were more frequently seen in children below the age of 5 than those above the age of 10. These changes were secondary to mechanical effects, which were seen clearly as mirror-image effects observed in compensatory curves. Here, posterior wedging was observed in 45 segments, as the compressive stress was applied more on the posterior aspects and tension forces on the anterior aspects of the vertebral endplates in the compensatory curves. According to the Hueter–Volkmann law, due to the increased forces acting in kyphosis, the growth and progression of deformities should slow down. However, Rajasekaran observed local growth acceleration, mainly in the front half of the boundary plate. A minimal increase in compressive forces could shift the Chondral Growth Force Response Curve to the right, but not so much as to exceed the “point of overload”. 

Anterior spine tethering aims to correct scoliotic deformities based on growth modulation due to the pressure exerted on the vertebral body epiphyseal growth plate [28]. The resumed activity of the vertebral growth plate was confirmed in our histological examination. Due to the lack of precise mathematical models, we decided to conduct experimental studies based on the rat tail spine [29]. Although Cheung et al. suggest that the possible etiologic factors causing scoliosis in lower animals are different from those in primates and that the findings of studies in lower animals cannot be applied to humans, their observations concerned the endocrine system [30]. On the other hand, the laws of physics that govern biomechanics are the same for all species. Originally, the mechanical stress caused changes in the growth plate, including changes in the width of the hypertrophic and proliferative zones and the final size of the hypertrophic chondrocytes. This is consistent with Stokes’ observations [31,32]. The high compressive force generated on the concave side inhibited the growth potential of this cartilage zone. The structures of the central and convex parts reacted similarly, though to a lesser extent. However, the width of the medial cartilage and the convex side did not differ statistically. This may mean that the unloading of the convex side did not increase the activity of the cartilage in this zone and, therefore, it does not fall under the Hueter–Volkmann law. 

Based on the previous considerations, the application of the compression force on the convex side of the deformation relieves the growth cartilage on the concave side and should shift the Chondral Growth Force Response Curve to the left, exceeding the “point of over-load”. The height of the chondrocytes along the entire width of the border plate increased and did not differ from the height of the chondrocytes in the control group (*p* > 0.05). The thickness of the growth cartilage on the concave side and in the centre of the border plate also returned to normal. On the convex side, it was even thicker than in the control group (*p* < 0.001). The height of the hypertrophic layer also increased after correction. The greatest increase in its thickness was noted on the concave side (nearly 2.4 times) (*p* < 0.001). However, it did not reach the thickness measured in the control group. 

It seems that among the changes in the structure of the growth plate mentioned above, the change in the final size of the hypertrophic chondrocytes is the most significant factor in the growth rate. Restoration of their natural height determines the proper function of the entire growth cartilage. Therefore, early correction of scoliosis and relief of the compressed growth plate can contribute to a return to the activity of the growth layer and hypertrophic chondrocytes as well as the repair and reconstruction of the vertebral body. 

Chay quantified changes in vertebral growth plates in response to asymmetrical compression via a unilateral corrective tether [33]. The researcher concluded that the anterior-based tether device achieved favourable three-dimensional correction versus the control group, without causing histological evidence of damage to the growth plates.

In our research, the influence of mechanical loads on the vertebral body is shown. Nevertheless, the effect of loads on the discs may be equally important [34,35]. Lalande et al. studied induced pressures on the epiphyseal growth plate with non-segmental anterior spine tethering. The author found a linear relationship between the applied tension and the resulting pressure, which remained similar from one level to another for a given configuration. An increased number of motion segments led to significantly reduced endplate pressures [36].

In clinical practice, correction is performed for the intervertebral disc where scoliosis has occurred. In severe scoliosis, structural changes in vertebral bodies are present due to growth disturbances of the concave side. Although correction is performed for the intervertebral disc, however wedged, the vertebral bodies of the concave side can be relieved and the endplate reactivated. This was observed in our experiment. That can lead to vertebral body remodelling and guarantee permanent stability.

A limitation of our study is the relatively small sample size despite compelling findings. The experiment was performed on an animal tail model, on a relatively stiff fixator different from tether cord. Larger validation studies can be envisioned for future research.

## 4. Materials and Methods

The study was approved by the Local Bioethics Committee for Animal Research (no. II/WL/1/2012). Twenty-four 49-day-old rats weighing 119–127 g (mean 125 g) were used in the study. The animals were caged with full access to standard laboratory chow and water in standard housing conditions (temperature 21 ± 1 °C, natural light/dark cycle). Surgical procedures were performed after three days of adaptation.

### 4.1. Method

The rat tail model was used in the study [37]. Asymmetric compression and distraction of caudal growth plates were performed using a modified external fixation apparatus developed by Mente et al. [38] based on the Ilizarov device. The apparatus was installed on the eighth and ninth caudal vertebrae under general anaesthesia (Ketamine 80 mg/kg, Xylazine 10 mg/kg). Buprenorphine 0.05 mg/kg was administered after the surgery. Four animals were sacrificed at the beginning of the experiment (Morbital 0.3 mL/kg) as a control group for the experiment (group I). In the first stage, the apparatus was calibrated to the pressure on the growth plate of 0.2 MPa (high compression according to Stokes [6]). The direction of the compressive force applied is shown in Figure 7a. After 3 weeks, eight rats were euthanised (Morbital 0.3 mL/kg), and material was collected for histopathological examination (animal age: 73 days, group II). In the second stage of the experiment, the compressive forces applied in the other animals (group III) were reversed (the rings of the apparatus were positioned parallel to achieve full correction) to obtain parallel distraction of the vertebrae (Figure 7b). After another three weeks of regeneration (animal age: 94 days), seven animals from group III were euthanised and material was collected for histopathological examination. In all groups, vertebrae without stabilisation were regarded as a control in the corresponding group. Five animals were excluded from the study: one animal due to improper insertion of the pins (group II animal), and two animals due to death during the apparatus insertion procedure (one animal each from groups II and III). In two rats, damage to the attachment of the wires to the ring occurred during the experiment (animals from group III).

### 4.2. Experimental Design

#### 4.2.1. Radiological Examination

The tail of each animal was examined radiographically in a controlled position relative to the film to ensure consistent and uniform magnification of images in the AP projection. Scans were performed at the beginning and end of each stage of the experiment. The angle of curvature was measured using Pdf-xchange viewer software version 2.5 from Tracker Software Products. The evaluation was carried out under a light microscope, and the digital image was analysed and measured in GNU Image Manipulation Program 2.8.2 for Linux.

#### 4.2.2. Histopathological Examination

The collected sections were decalcified and embedded in paraffin blocks. The slides were cut in the coronal plane and stained with haematoxylin and eosin and Methyl Blue. They were evaluated under a light microscope, and the digital image was analysed and measured in GNU Image Manipulation Program 2.8.2 for Linux.

In the control vertebrae, the measurements of the thickness of the growth plate, hypertrophic layer, and chondrocytes were averaged over the total vertebral body width. 

The endplate in the deformed vertebrae was divided into three equal parts and labelled as the concave, central, and convex sides. The division and photograph of a vertebra X-ray image in each group are shown in Figure 8. 

Three measurement areas were distinguished in each part. The upper endplate of the hypertrophic layer was identified as the first chondrocyte layer. The lower endplate was constituted by the farthest intact chondrocytes on the epiphysis. After determination of nodal points, curves connecting these points were created. The height of the hypertrophic zone was measured as the distance between the points of the curve along the long axis line perpendicular to the epiphysis. The scheme of the hypertrophic zone measurement is shown in Figure 9.

The height of the chondrocytes was measured between the upper and lower margins of a single cell. The chondrocyte height was the mean height of all cells measured in the section. The scheme of the chondrocyte height measurement is shown in Figure 10.

### 4.3. Statistical Analysis 

Statistica ver. 13.3 (TIBCO Software Inc., Paol Alto, CA, USA, 2017) was used to perform statistical analyses. The data were analysed with one-way analysis of variance, and the post hoc Tukey test (*p* < 0.05) was applied for estimation of the significance of the differences between the treatments. The data on the growth plate thickness, hypertrophic layer thickness, and chondrocyte height were subjected to principal component analysis (PCA).

## 5. Conclusions

The results of this study indicate that mechanical modulation can be used to reverse deformation of the vertebral body. Initiation of the correction of scoliosis in the growing spine, with relief of the pressure on the growth plate, allows the return of the physiological activity of the growth cartilage and the restoration of the deformed vertebral body. Compression of the convex side does not change the activity of the corresponding part of the growth plate.

## Figures and Tables

**Figure 1 ijms-23-11596-f001:**
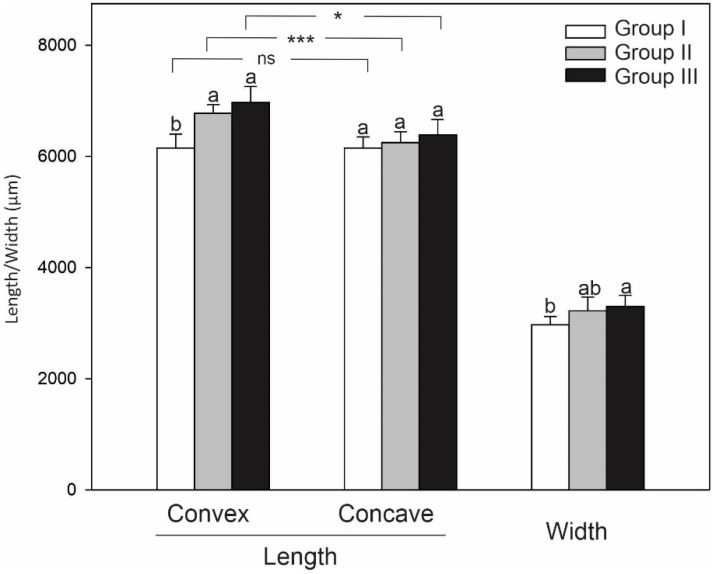
Mean (±SD) morphometric parameters of vertebrae in three groups of treatment: group I, beginning of experiment; group II, induced scoliosis; group III, after correction. Values followed by different letters are significantly different according to Tukey’s test (*p* < 0.05); * and *** indicate significant differences between convex and concave length within the same group at the 0.05 and 0.001 levels of Student’s *t*-test; ns, non-significant.

**Figure 2 ijms-23-11596-f002:**
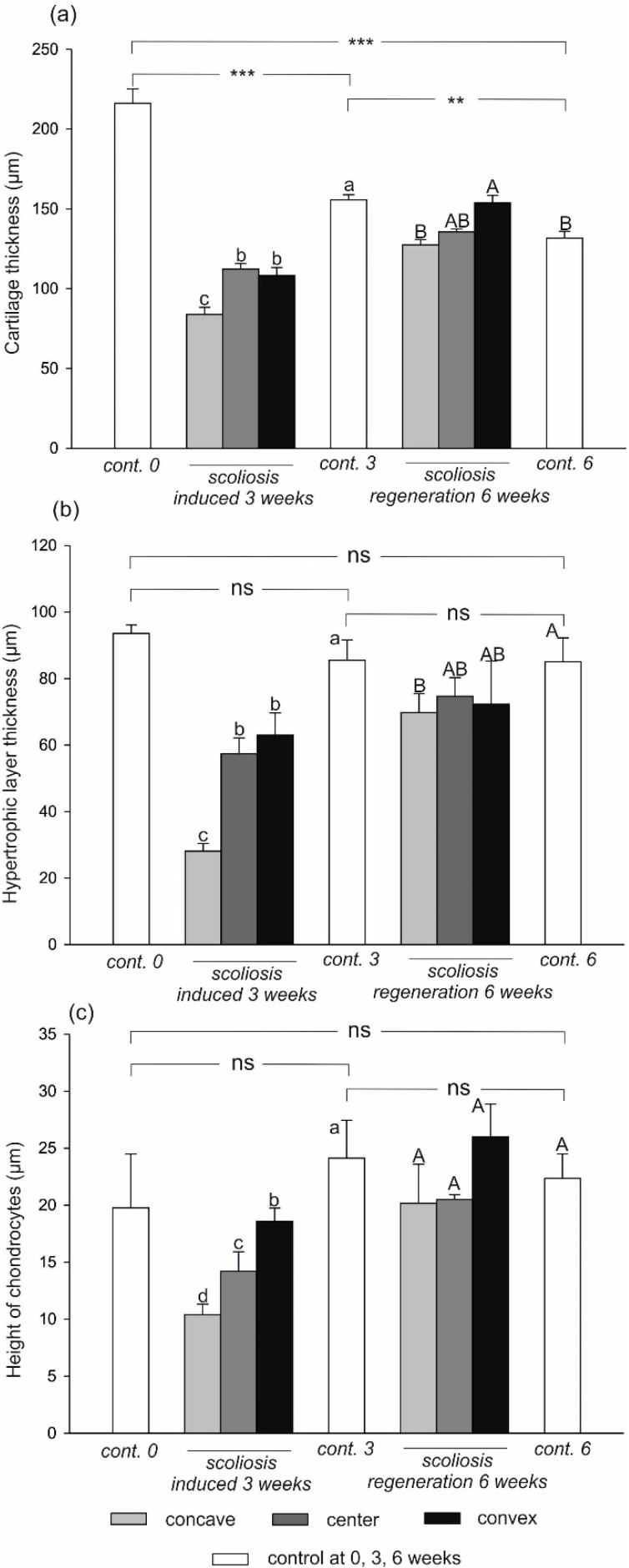
Cartilage thickness (**a**), hypertrophic layer thickness (**b**), and height of chondrocytes (**c**) in the rat treatments. Data are means ± SD (n = 4). Means followed by various letters differ significantly (*p* < 0.05, Tukey’s test) **, and *** indicate significant differences from the different controls at the, 0.01, and 0.001 levels of Student’s *t*-test at 0, 3, and 6 weeks.

**Figure 3 ijms-23-11596-f003:**
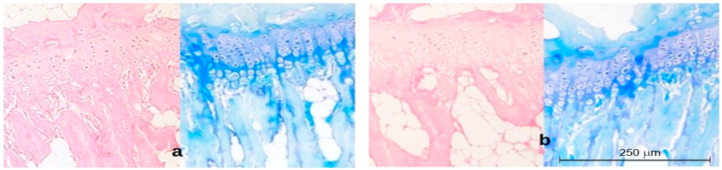
Histopathological image of cartilage in the control group of 52-day-old animals (day of the start of the experiment) (group I) (**a**) and the control group of 94-day-old animals (day of the end of the experiment) (group III) (**b**). H&E and Methyl Blue staining 200×.

**Figure 4 ijms-23-11596-f004:**
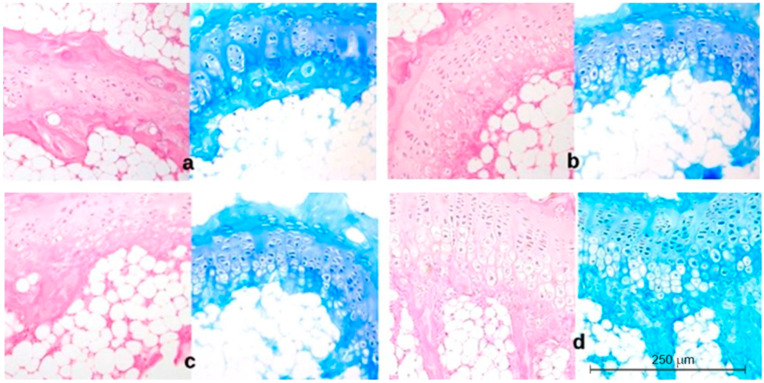
Group II. Histopathological image of cartilage from the concave side (**a**): irregular arrangement of the columns of the proliferative layer, thin growth cartilage, atrophic hypertrophic zone; central side (**b**): regular arrangement of the columns of the proliferative and hypertrophic layers; convex side (**c**): less regular arrangement of the columns of the proliferative and hypertrophic layers than in the control group, active hypertrophic layer; control group (**d**): normal and regular arrangement of the columns of the proliferative and hypertrophic layer, wide growth cartilage, fully active hypertrophic layer. Seventy-three-day-old animals (completion of scoliosis induction); H&E and Methyl Blue staining 200×.

**Figure 5 ijms-23-11596-f005:**
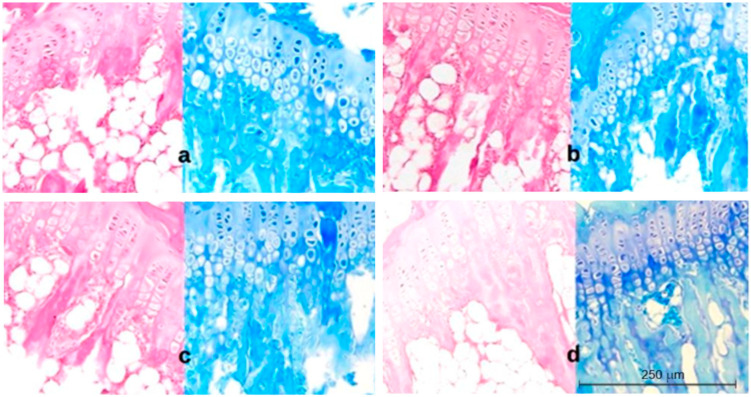
Group III. Histopathological image of cartilage from the concave side (**a**): regained regular arrangement of the proliferative and hypertrophic layers, reactivation of the hypertrophic layer; central side (**b**): regular arrangement of the columns of cells, wide hypertrophic layer; convex side (**c**): regular arrangement of the columns of cells, wide hypertrophic layer; control group (**d**): regular arrangement of the columns of cells, active hypertrophic layer. Ninety-four-day-old animals (end of the experiment after regeneration); H&E and Methyl Blue staining 200×.

**Figure 6 ijms-23-11596-f006:**
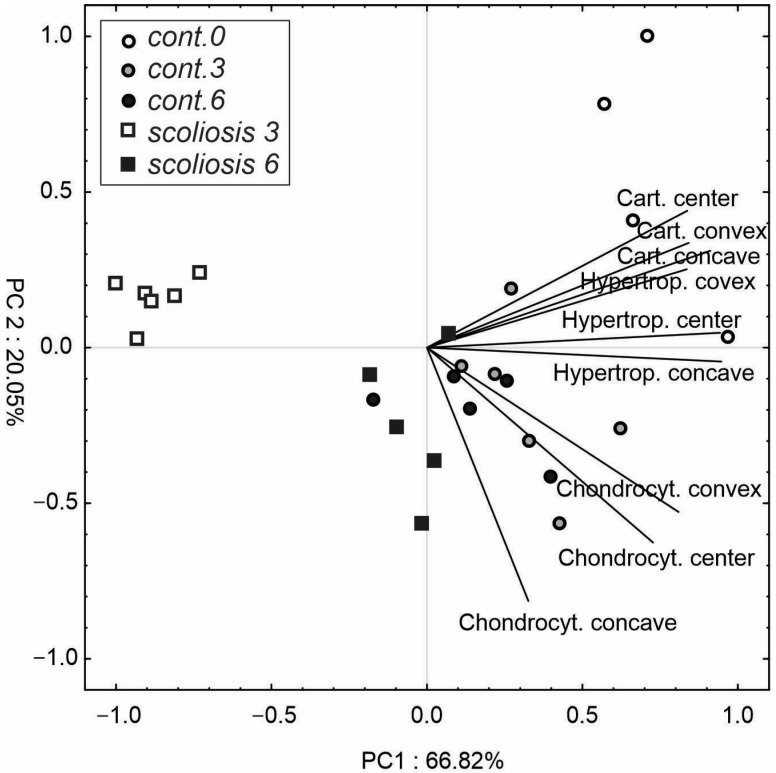
Principal component analysis (PCA) of variables (cartilage thickness, hypertrophic layer thickness, and height of chondrocytes) in the different treatments. The length of the lines shows a correlation between the original data and the PC axes.

**Figure 7 ijms-23-11596-f007:**
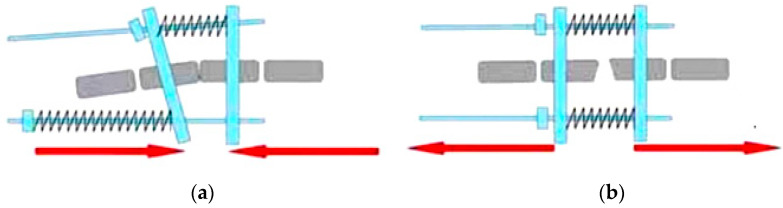
Scheme of forces applied during the experiment: (**a**) compression, group II; (**b**) distraction, group III.

**Figure 8 ijms-23-11596-f008:**
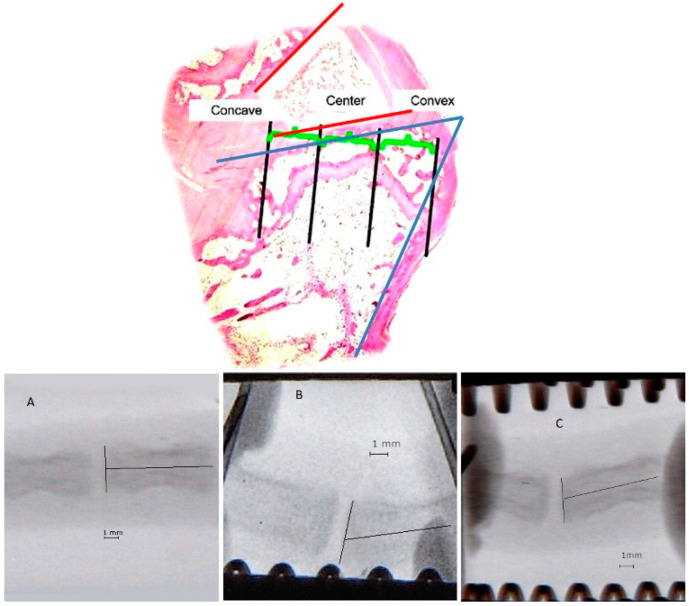
Photograph of a histological slide of the endplate divided into three parts: concave, central, and convex. Visible wedging of the vertebra (blue angle) and deformed disc (red angle). H&E 20× staining. Photograph of a vertebra X-ray image in group I (**A**), group II (**B**), and group III (**C**).

**Figure 9 ijms-23-11596-f009:**
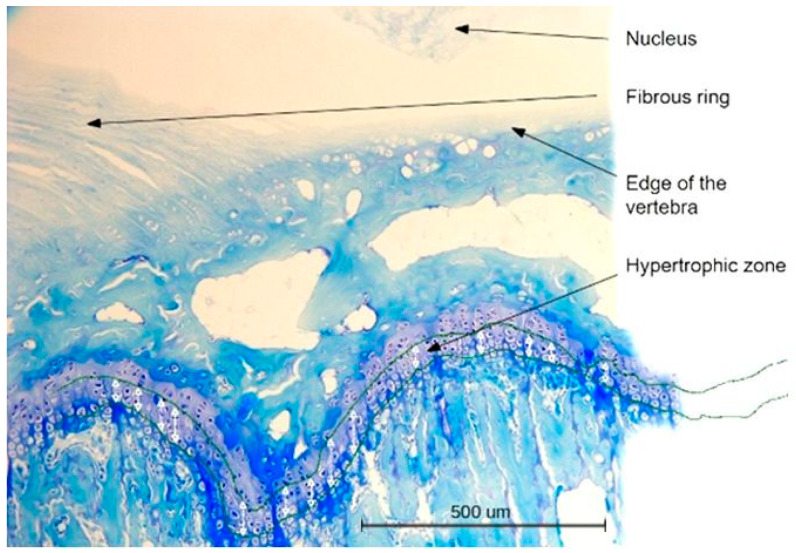
Scheme of hypertrophic zone measurement. Methyl Blue staining 100×.

**Figure 10 ijms-23-11596-f010:**
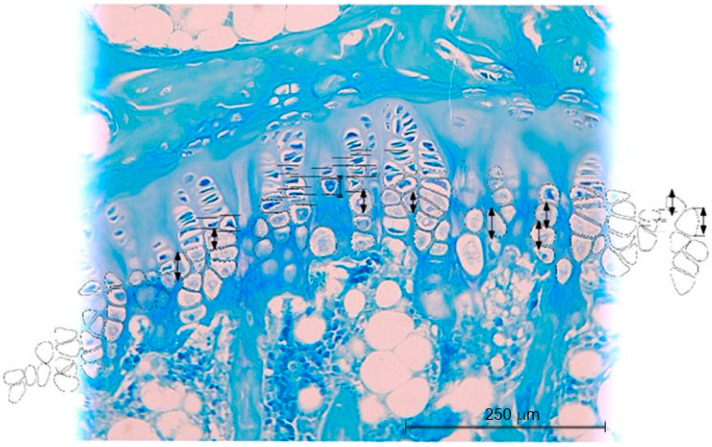
Scheme of chondrocyte height measurement. Methyl Blue staining 200×.

## Data Availability

The data presented in this study are available on request from the corresponding author.

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
