# Peer review of "Reactivation of Vertebral Growth Plate Function in Vertebral Body Tethering in an Animal Model"

_ijms, 2022, doi:10.3390/ijms231911596_

Round 1
Reviewer 1 Report
This manuscript describes the basic study regarding scoliosis surgery without rigid fusion. Although it appears to be of some value to the reader, this is stylistically not at the level of publication.
1. Methods section is after Discussion session and there are two types of Figures 1-4 (page 3-5 and page 9-11). It's funny that all eight authors don't realize it. Do co-authors really check final submissions? No careless mistakes are allowed. The author should know that your credit has been impaired.
2. There is no scale in the photos of histological section without Figure 3 in page 11.
3. The characters in the pictures of Figures 2, 3, and 4 (page 4 and 5)are unnecessary. Should be listed as Legend.
4. This is a study on the endplate of local scoliosis artificially created by external fixation. However, in clinical practice, correction is performed for the intervertebral disc where scoliosis has occurred, and in the present experimental model, we did not see changes in the endplates when correcting scoliosis. I think it should be stated in the discussion, but what do you think?
Author Response
Reviewer 1
Dear Reviewer,
Thank You very much for the effort you made to review our manuscript and for valuable suggestions to allow us improve the paper. Below we attach, the response point by point and all corrections in the text have been marked on red. We hope that our explanations are exhausting.
- Methods section is after Discussion session and there are two types of Figures 1-4 (page 3-5 and page 9-11). It's funny that all eight authors don't realize it. Do co-authors really check final submissions? No careless mistakes are allowed. The author should know that your credit has been impaired.
We deeply apologies for our mistake, the numbering of figures has been corrected.
- There is no scale in the photos of histological section without Figure 3 in page 11.
Thank you for your comment. It has been corrected.
- The characters in the pictures of Figures 2, 3, and 4 (page 4 and 5)are unnecessary. Should be listed as Legend.
It has been corrected according your suggestion.
- This is a study on the endplate of local scoliosis artificially created by external fixation. However, in clinical practice, correction is performed for the intervertebral disc where scoliosis has occurred, and in the present experimental model, we did not see changes in the endplates when correcting scoliosis. I think it should be stated in the discussion, but what do you think?
Thank you for your comment. You are right that in clinical practice, correction is performed for the intervertebral disc where scoliosis has occurred. However in severe scoliosis, where structural changes are present, correction is performed for the intervertebral disc but wedged, concave side’s end plate can be reactivated. It was observed in our experiment. This can lead to vertebral body remodeling which can guarantee permanent stability. – the appropriate comment has been added to discussion (Line 279-284).
Reviewer 2 Report
This article covers an interesting area of research. It shows that initiation of correction of scoliosis in the growing spine with relieving pressure through mechanical modulation on the growth plate promises the return of the physiological activity of the growth cartilage and restoration the deformation of the deformed vertebral body. This work assessed and proved the reactivation of vertebral growth plate function after applying forces correcting deformation. There is only some concern.
1, There should be X-ray data of the vertebrae of each group, which displays the morphology of the vertebral, to prove achieved deformation and restoration of the vertebral. Only HE and toluidine blue staining cannot confirm the overall morphology of the vertebrae.
2, “In the second stage of the experiment, the compressive forces applied in the other animals (group III) were reversed to obtain parallel distraction of the vertebrae”. How much force of reversed parallel distraction is applied in animals of group III? Or is it randomly set?
3, The return of the physiological activity of the growth cartilage and restoration of the deformed vertebral body were proved by H&E and Methyl Blue staining. However there is no data that proves the endochondral ossification in this article. And Movat pentachrome staining may be the suitable experimental method for displaying mineralized tissue, cartilage, muscle and bone marrow.
4, Recent literature rather than previous literature should be cited to demonstrate recent advances in the field.
Author Response
Reviewer 2
Dear Reviewer,
Thank You very much for the effort you made to review our manuscript and for valuable suggestions to allow us improve the paper. Below we attach, the response point by point and all corrections in the text have been marked on red. We hope that our explanations are exhausting.
1, There should be X-ray data of the vertebrae of each group, which displays the morphology of the vertebral, to prove achieved deformation and restoration of the vertebral. Only HE and toluidine blue staining cannot confirm the overall morphology of the vertebrae.
Thank you for your comment, we agree with your opinion that the staining cannot confirm the morphology of the vertebrae. Morphology of the vertebrae was assessed by radiological measurements. Detailed data has been presented in form of additional figure (Fig. 1) to show changes of the vertebrae in each group. The significant difference (p= 0.001) between convex and concave of the vertebrae in group II proved vertebrae was deformed; in turn, the decreased difference between convex and concave of the vertebrae in group III showed that vertebrae was restored.
2, “In the second stage of the experiment, the compressive forces applied in the other animals (group III) were reversed to obtain parallel distraction of the vertebrae”. How much force of reversed parallel distraction is applied in animals of group III? Or is it randomly set?
Thank you for your comment, the information has been added to the text: “In the second stage of the experiment, the compressive forces applied in the other animals (group III) were reversed (the rings of the apparatus were positioned parallel to achieve full correction) to obtain parallel distraction of the vertebrae (Fig 6 b).”
3, The return of the physiological activity of the growth cartilage and restoration of the deformed vertebral body were proved by H&E and Methyl Blue staining. However there is no data that proves the endochondral ossification in this article. And Movat pentachrome staining may be the suitable experimental method for displaying mineralized tissue, cartilage, muscle and bone marrow.
Thank you for your valuable comment. Our experiment was finished quite a long time ago and unfortunately we have no material to carry out Movat pentachrome staining. However, thank You – it is valuable suggestion for our next project. H&E and Methyl Blue staining were enough to prove the return of the physiological activity of the growth cartilage what was the aim of this studied. The Movat pentachrome staining is planned for the next investigation focused on mineralized tissue, cartilage, muscle and bone marrow.
4, Recent literature rather than previous literature should be cited to demonstrate recent advances in the field.
Some recent literature has been added to manuscript according to your suggestion. However some older references have been retained because they presented theoretical models and principal rules e.g. the biology of bone growth.
Round 2
Reviewer 2 Report
This article shows that initiation of correction of scoliosis in the growing spine with relieving pressure through mechanical modulation on the growth plate promises the return of the physiological activity of the growth cartilage. In the revised manuscript, authors did make a lot of improvements, but there are still some problems.
1, “Mean (+- SD)” in the legend of figure 1, is the careless mistake.
2, The sizes of the graphs in figure 5 are inconsistent.
3, It's not “H+E”, but “H&E”. Please check final submissions carefully.
4, There should be original radiograph of the vertebrae of each group, which displays the morphology of the vertebral, to prove achieved deformation and restoration of the vertebral.
Author Response
Dear Reviewer,
Thank You very much for all suggestions to allow us improve the paper. Below we attach, the response point by point and all corrections in the text have been marked on red.
1, “Mean (+- SD)” in the legend of figure 1, is the careless mistake.
It has been corrected. Thank You.
2, The sizes of the graphs in figure 5 are inconsistent.
The sizes of the graphs have been corrected.
3, It's not “H+E”, but “H&E”. Please check final submissions carefully.
It has been corrected. Thank You.
4, There should be original radiograph of the vertebrae of each group, which displays the morphology of the vertebral, to prove achieved deformation and restoration of the vertebral.
The example of original radiograph of the vertebrae in each group has been added as a part of figure 8.